# Regenerative Capacity of Endogenous Factor: Growth Differentiation Factor 11; a New Approach of the Management of Age-Related Cardiovascular Events

**DOI:** 10.3390/ijms19123998

**Published:** 2018-12-12

**Authors:** Luc Rochette, Alexandre Meloux, Eve Rigal, Marianne Zeller, Yves Cottin, Gabriel Malka, Catherine Vergely

**Affiliations:** 1Equipe d’Accueil (EA 7460): Physiopathologie et Epidémiologie Cérébro-Cardiovasculaires (PEC2), Université de Bourgogne—Franche Comté, Faculté des Sciences de Santé, 7 Bd Jeanne d’Arc, 21000 Dijon, France; alexandre.meloux@gmail.com (A.M.); eve.rigal@u-bourgogne.fr (E.R.); marianne.zeller@u-bourgogne.fr (M.Z.); yves.cottin@chu-dijon.fr (Y.C.); catherine.vergely@u-bourgogne.fr (C.V.); 2Service de Cardiologie—CHU-Dijon, 21 000 Dijon, France; 3Institut de formation en biotechnologie et ingénierie biomédicale (IFR2B), Université Mohammed VI Polytechnique, 43 150 Ben-Guerir, Morocco; catherine.vergely@u-bourgogne.fr

**Keywords:** GDF11, regenerative, parabiosis, cardiovascular events, ageing

## Abstract

Aging is a complicated pathophysiological process accompanied by a wide array of biological adaptations. The physiological deterioration correlates with the reduced regenerative capacity of tissues. The rejuvenation of tissue regeneration in aging organisms has also been observed after heterochronic parabiosis. With this model, it has been shown that exposure to young blood can rejuvenate the regenerative capacity of peripheral tissues and brain in aged animals. An endogenous compound called growth differentiation factor 11 (GDF11) is a circulating negative regulator of cardiac hypertrophy, suggesting that raising GDF11 levels could potentially treat or prevent cardiac diseases. The protein GDF11 is found in humans as well as animals. The existence of endogenous regulators of regenerative capacity, such as GDF11, in peripheral tissues and brain has now been demonstrated. It will be important to investigate the mechanisms with therapeutic promise that induce the regenerative effects of GDF11 for a variety of age-related diseases.

## 1. Introduction

The prevalence of a number of chronic diseases is much higher in the elderly. These diseases include cardiovascular diseases, stroke, cancer, and dementia. There are strong similarities between the cellular changes related to aging and those found in some of the most common chronic, non-communicable, and mostly multifactorial diseases. Several major theories have been put forward to explain the biology of aging, and these are not mutually exclusive. It is now well established that cellular senescence is associated with the activation of pro-inflammatory pathways and free radical production [1]. Novel ways to diagnose aging involving telomeres and other markers of senescence are being validated. Biological factors appear as “mediators,” which can be targeted to slow down or even reverse disease or physiological conditions such as aging [2]. It is widely accepted that the rate of aging reflects the ratio of tissue attrition to tissue regeneration; it thus seems credible to assume that accelerated aging can result from an alteration of this balance, caused by either the exacerbation of tissue attrition or a defect in tissue regeneration. The human heart is a highly dynamic organ that retains a significant degree of plasticity throughout life and even in the presence of heart failure. The reversibility of certain aspects of aging has become increasingly appreciated. The stem cell compartment appears to be properly equipped to regulate tissue homeostasis, but this function is determined by the plasticity and heterogeneity of adult stem cells [3,4,5].

## 2. Mechanistic and Evolutionary Bases of Aging

Aging is characterized by a regular functional drop of all organs. It is endogenous, progressive, and deleterious for the individual. Among all aging theories, one describes nitro-oxidative stress associated with the overproduction of reactive oxygen species (ROS) and reactive nitrogen species (RNS). It implies that progressive aging is associated with metabolic diseases [6,7].

Vascular aging is the major contributing factor to increases in the incidence and prevalence of cardiovascular disease, due mainly to the presence of chronic, low-grade inflammation. Aging produces its own cardiovascular changes, mainly remodeling of arteries and the myocardium [8]. Cardiovascular diseases (CVD), including hypertension, coronary artery disease (CAD), atrial fibrillation and chronic heart failure (CHF), are highly prevalent in the elderly. CHF associated with hypertrophy is a mounting health problem that affects up to 10% of people over the age of 65 and it is associated with frequent hospital admissions, reduced quality of life, significant morbidity, and increased mortality. Cardiac hypertrophy is considered an initially adaptive response that counteracts increased wall tension and helps maintain cardiac output. However, if the heart is persistently exposed to increased loads, cardiac hypertrophy may become maladaptive, leading to heart failure. Aging is associated with an extensive loss of function at all levels of biological organization. Studies using model organisms have generated significant insights into the genetic factors and environmental conditions that influence the age-related decline [9,10]. Progressive aging induces several structural and functional alterations in the cardiovascular system, among which a reduced number of myocardial cells and increased interstitial collagen fibers are particularly important as they result in the development of heart failure. The signals and mechanisms that cause age-related tissue failure are unclear. One of these alterations is a change in the activity of the protein kinase Akt, which plays a central role in regulating a variety of cellular processes ranging from cell survival to aging. Extensive studies have established that growth factors, hormones, and cytokines stimulate Akt activation through serial phosphorylation events. Accumulating evidence illustrates that different Akt isoforms are responsible for their diverse biological functions.

In cardiac hypertrophy and aging, sirtuins (SIRT) have been shown to regulate Akt signaling. The SIRT isoforms SIRT 1, SIRT3, and SIRT6 play a central role in the regulation of Akt activation. Although SIRT1 deacetylates Akt to promote phosphatidylinositol (3,4,5)-triphosphate binding and activation, SIRT3 controls ROS-mediated Akt activation, and SIRT6 transcriptionally represses Akt at the level of chromatin [11]. In the cardiovascular field, treatments can protect heart muscle tissue after acute myocardial infarction (MI). Post-infarction remodeling and the evolution to CHF remain a problem in the treatment of cardiovascular disorders. Tissue regeneration is characterized by composite drops of growth factors with decisive roles in cell proliferation and differentiation. [12]. Numerous trials have been started to explore the transplantation of stem-cell populations for cardiac regeneration. Although most cell types produced hopeful results in vitro and in preclinical studies, they were disappointing in terms of clinical profits [13,14].

Growth differentiation factor 11 (GDF11) belongs to the transforming growth factor β: TGF-β family. Members of the TGF-β superfamily are highly conserved across animals. They are ubiquitously expressed in diverse tissues and function during the earliest stages of development and throughout life. The TGF-β superfamily of secreted factors comprises more than 30 members including activins, nodals, bone morphogenetic proteins (BMPs), and the GDFs such as GDF11 and GDF15. TGF-β family members are involved in a wide range of functions and play key roles in embryogenesis, development, and tissue homeostasis.

GDF11, like other members of the TGF-β superfamily, are generated from precursor proteins by proteolytic processing [15]. After cleavage of a single peptide bond by a furin-type protease, the N-terminal propeptide and the disulfide-bonded homodimer containing the mature growth factor domains remain connected, establishing an inactive and latent complex. The active mature growth factors may be released from the latent complexes beyond degradation of the propeptide via proteases [16] (Figure 1). GDF11 is a disulfide-linked dimer. It is firstly synthesized as precursor and next cleaved by furin-like proteases separating the N-terminal prodomain from the C-terminal domain.

GDF11 is expressed in the pancreas, intestine, kidney, skeletal muscle, heart, developing nervous system, olfactory system, and retina [17]. Its expression is present in young adult organs and decline during aging. In return, a study [18] demonstrated that GDF11 serum levels do not reduce during aging. The reagents used in some studies were not GDF11-specific, interfering with myostatin evaluation. The amino acid sequence of GDF11 is 90% identical to that of myostatin [19]. Recently, based on quantification by Western blot analysis, a study showed that circulating GDF11/8 levels decreased with age in mice as well as in other mammalian species, and that increasing GDF11/8 levels with exogenous GDF11 regulated cardiomyocyte size [20].

Clearly, the search for humoral factors, such as GDF11, able to affect regenerative capacity, should be re-examined [21]. Different functions of GDF11 in controlling progenitor proliferation and/or differentiation were demonstrated [22]. Experimental methods using techniques of parabiosis demonstrate that impaired regeneration in aged mice is reversible by exposure to circulation of young animals. GDF11 controls heart and muscle aging, suggesting that young blood contains “rejuvenating” factors that can renovate regenerative process.

## 3. The Surgical Technique of Parabiosis

In order to investigate the influences of an organism on its conjoined partner, animal models that “mimic” the natural phenomenon of conjoined twins were created. The surgical technique of physically connecting two living organisms was termed “parabiosis”, from the Greek “para” (next to) and “bios” (life). Unlike transbiosis, there are no formal donors and hosts in parabiosis as each animal can be viewed as an equal partner in the pairing, each influencing the other animal called a parabiont. These methods have evolved over time and have involved the surgical connection of different body parts [23]. Use of the parabiotic model in studies of cutaneous wound healing to define the participation of circulating cells [24,25].

The technique was first introduced by the French physiologist Paul Bert in the 1860s. The surgical way involves skin incisions along the contiguous flanks of two animals (usually mice) and suturing adjacent skin flaps between the animals. The peritoneum is sutured together between the animals to form a common peritoneal cavity. The first studies started to graft animals of different ages to each other (heterochronic parabiosis (HP)) in order to investigate effects induced through exposure of an aged organism to a youthful systemic environment. In HP, the parabiotic pairing of two animals is of different ages. Parabiotic pairings between two young (Y) mice or two old (O) mice are isochronic parabiosis (IP). It is an experimental system to test for systemic effects on the process of tissue aging. Collecting verification has indicated that the blood of young animals holds powerful “factors of youth”.

## 4. Does GDF11 Reverse Age-Related Functional Impairment in Heart?

As we reported previously, cardiac hypertrophy is a prominent pathological feature of age-related heart failure. Using the parabiosis model, it has been demonstrated that age-related cardiac hypertrophy can be reversed via exposure to a young circulatory environment. These experiments revealed that age-related cardiac hypertrophy is at least in part mediated by circulating factors, such as GDF11, which is able to reverse the condition [26]. The authors generated HP pairs, in which Y female C57BL/6 mice (Y-HP, 2 months) were surgically joined to O-partners (O-HP, 23 months), and compared these with IP pairs (Y-Y, Y-IP, or O–O, O-IP), joined at identical ages, and to age- and sex-matched unpaired mice as controls. Parabiotic pairs were continued for 4 weeks before analysis. The hearts from old mice exposed to a young circulation (O-HP) for these 4 weeks were patently smaller than hearts from O-IP mice. The reduction of cardiac hypertrophy was associated with changes in cellular hypertrophy. This effect is evaluated by morphometric analysis of cardiac histologic sections. The reversal of cardiac hypertrophy in old mice exposed to a young circulation cannot be explained by a reduction in blood pressure in the older mice. An extensive proteomics analysis was performed on the serum and plasma of the animals. GDF11 was reduced in the circulation of aged mice and its levels were restored to those in young animals by HP. A significant decrease was also found in both GDF11 gene expression and GDF11 protein levels in the spleens of old mice. These results suggest exciting therapeutic approaches for the management of age-related cardiac hypertrophy by restoring youthful levels of circulating GDF11. Much work remains to be done to establish the efficacy of GDF11 as a therapeutic rejuvenation factor [27]. Recently, the goal of a study in old mice [28] was to reexamine the possibility to restore youthful levels of GDF11 by injecting rGDF11 and thus reversing cardiac hypertrophy and imparting a young phenotype to the old heart. The conclusions were that recombinant GDF11 (rGDF11) had no effect on cardiac structure and cardiac pump function; these results do not support the concept that GDF11 could be an anti-aging compound. An important study was published in the clinical cardiovascular field, analyzing the Heart and Soul and HUNT3 cohorts. It was demonstrated, in patients with stable ischemic heart disease, that higher GDF11/8 levels were associated with a lower risk of cardiovascular events and death, suggesting cardioprotective properties of GDF11/8 [29].

Various studies have identified a second factor in blood from mice that mediates rejuvenating actions, a chemokine: C-C motif chemokine 1, CCL11 [30]. CCL11 is also known as eosinophil chemotactic protein. In contrast to GDF11, levels of CCL11 increase with age (Figure 2). Increased circulating levels of CCL11 have been implicated in diseases associated with abnormal central nervous system function, including Alzheimer disease [31].

In the cardiovascular field, it is important to note the complex interplay of various cytokines in maintaining normal hematopoiesis. The process of differentiation of hematopoietic stem cells into mature blood cells is tightly regulated by the actions of both stimulatory and inhibitory cytokines. Recent studies suggest a new pathway for the treatment of anemia by targeting a newly discovered regulator of erythropoiesis via GDF11 [32].

GDF11 (active form: 407 amino-acids) exerts its effects by binding to specific receptors. Members of the TGF-β family bind to acting receptor-like-kinases: ALK-Type I and Type II. ALK-I induces phosphorylation of Smads (Figure 1). Studies reported that GDF11 Activin-type II receptors B(ActRIIB)-Smad2/3-dependent signaling is a key regulatory mechanism in proliferating erythroid precursors as it controls their late-stage maturation. New observations suggest that ActRIIA ligand traps may have therapeutic relevance in diseases such as beta-thalassemia by suppressing the deleterious effects of GDF11 [33]. The cycle is associated with changes in iron utilization caused by the decreased expression of hepcidin, the hormone that regulates iron uptake [34].

## 5. Does GDF11 Reverse Age-Related Functional Impairment in Skeletal Muscle?

Evidence indicates that skeletal muscle influences systemic aging. An age-dependent decline in skeletal muscle mass, strength, and endurance during the aging process is a physiological development and several factors may exacerbate this process. Capillaries are an integral part of the mechanism underlying this close matching between blood flow and metabolism of skeletal muscle mass. Capillaries have the capacity to play an active role in co-ordination of muscle blood flow responses to changed muscle metabolism [35].

Adult skeletal muscle possesses extraordinary regeneration capacities. Muscle satellite cells are responsible for the postnatal growth and major regeneration capacity of adult skeletal muscle. Aged serum increased the myogenic-to-fibrogenic conversion of young cells, whereas young serum had the opposite effect on aged cells [36]. The number of satellite cells is decreased during ageing, resulting in the attenuation of muscle regeneration capacity [37]. Several growth factors and cytokines such as TGF-β, and myostatin regulate muscle growth and repair [38,39]. Previous studies demonstrated that impaired regeneration in aged muscle can be reversed by HP, which exposes aged tissues to a youthful systemic environment and restores injury-induced satellite cell activation by the up-regulation of Notch signaling [40]. To determine whether supplementation of GDF11 from the young partner might underlie changes in skeletal muscle in heterochronic parabionts, aged mice were treated with daily intraperitoneal injections of rGDF11 to increase systemic GDF11 levels [41]. After 4 weeks, satellite cell frequency, determined by flow cytometry, and function increased in the muscles of rGDF11-treated mice, whereas other myofiber-associated mononuclear cell populations were unaffected. Aged mice treated with rGDF11 also showed increased numbers of satellite cells with intact DNA. These results indicate that GDF11 is able to regulate muscle aging and may be therapeutically suitable for skeletal muscle dysfunction.

## 6. Does GDF11 Reverse Age-Related Functional Impairment in Brain? Importance of the Neurovascular Unit

Normal human aging is a complex biological process that results from gradual phenotypic and functional changes of cells, which may be influenced by changes in their tissue microenvironment that occur with time. It has been demonstrated that alterations in brain structure and function are intimately tied to alterations in cognitive function. Age-related mechanisms impair the function of cells of the neurovascular unit. The brain must rely on the circulation for continuous supply of nutrients and oxygen, and for liberating metabolic spare products [42].

In humans and mice, the hippocampus and the hypothalamus are particularly vulnerable to aging. In addition to age-related differences in white and gray matter integrity, there is some evidence that age is associated with changes in the vascular system [43,44]. Older adults often have less activity in some regions, such as medial temporal areas during memory processing and visual regions across a variety of cognitive domains. A growing body of evidence shows that changes in the phenotypic and functional properties of human adult stem/progenitor cells may occur during aging and are associated with pathological consequences. Regeneration activity declines with age with the loss of stem/progenitor cell function. The brain is one of the organs that harbor a very small subpopulation of adult stem/progenitor cells, corresponding to about 0.1–3% of the total cell mass. Molecular mechanisms that may contribute to extending the longevity of adult stem/progenitor cells include their high telomerase activity, slow division, and high resistance to internal and external injuries [45].

Novel therapeutic approaches with multiple applications in humans have been conceived to reverse this chronological decline. These approaches include therapeutic strategies such as gene therapies and cell-replacement by using functional adult stem/progenitor cells [46]. Many pro-longevity signaling pathways, such as class O of forkhead box (Foxo) transcription factors and SIRT1, have been shown to play important roles in brain function. The manipulation of signaling molecules that impact Foxo and SIRT1 activities improved neuronal stress response capabilities to ROS and extended overall lifespan [47].

As we noted previously, several studies have reported that aged mice exposed to a young systemic environment exhibited reduced signs of biological aging in cardiovascular and skeletal muscle systems. The rejuvenating effects of young blood have now also been observed in the central nervous system of old mice. The rejuvenation of brain regeneration in aging organisms has been observed after HP. It has been reported that blood-borne factors, such as the chemokine CCL11, present in the systemic milieu were able to inhibit or promote adult neurogenesis in an age-dependent fashion in mice [48]. In a mouse model of demyelination, exposure of old mice to a youthful systemic environment increased myelination in the spinal cord of old heterochronic parabionts by recruiting young peripheral monocytes and promoting differentiation of oligodendrocyte progenitor cells [49]. Research has been done on whether or not the age-related decline in neurogenic function could be restored by extrinsic young signals, using a mouse heterochronic parabiosis model. It was demonstrated that aged cerebral vasculature was remodeled in response to young systemic factors, producing noticeably greater blood flow, associated with an activation of subventricular zone neural stem cell proliferation and enhanced olfactory neurogenesis, leading to an improvement in olfactory function. GDF11 was able to increase blood flow and neurogenesis in aged mice. In the context of aging, GDF11 appears to promote plasticity of the central nervous system [50]. It has been reported that old mice that received repeated injections of plasma from young animals exhibited increased learning and memory compared with old mice that received injections of old plasma. Structural and cognitive enhancements elicited by exposure to young blood are mediated, in part, by activation of the cyclic AMP response element binding protein in the aged hippocampus [30].

Cell culture experiments suggest that neurogenic rejuvenation is due to rGDF11-induced activation of the TGF-β signaling pathway in brain capillary endothelial cells, thus increasing their proliferation [50]. Ligand-receptor interactions in the TGF-β superfamily have been extensively studied. Biochemical experiments have demonstrated that GDF11 can activate Smad2, suggesting the involvement of ALK receptors in GDF11 signaling (Figure 1). It is important to recall that signaling through activin receptors is a therapeutic target in multiple diseases [51].

As previously reported, neurons, blood vessels, and neuroglia constitute the neurovascular unit. This unit appears fundamental for the brain metabolism and is regulated by numerous endogenous modulators. Recent evidence supports the hypothesis that, during inflammation, TGF- β can have an effect on the brain-wide pathway for fluid transport in the brain. The inhibition of inflammation appears to be a potential therapeutic strategy for neuro-inflammatory injury [52]. In this context, the role of GDF11 may be evoked; new investigations are required to determine the specific actions of GDF11 on the neurovascular unit and the blood–brain barrier permeability. 

## 7. Limits and Criticism Concerning the Properties of GDF11: What is the Specificity of the Actions of GDF11?

As we reported, there is a high degree of identity between the GDF8/myostatin and GDF11 active domains. Like myostatin, the GDF11 protein is detectable in human serum [29] and signals through the activating receptors. GDF11 and GDF8 activate the same receptors. ActRIIB mediates multiple signals for TGF-beta family members, including activin, GDF8, and GDF11 [53]. As GDF11 and GDF8 are highly homologous, they are difficult to distinguish biochemically and by most commercial antibodies. When systemic GDF11 levels were evaluated without myostatin cross-reactivity, they were found to be 500× lower than those of myostatin. It has been suggested that reagents used in some studies were inadequate to determine whether GDF11 levels in old mice fell or increased after rGDF11 injection. It is important to use an assay with appropriate sensitivity and specificity for GDF11 detection [54].

A component of understanding the functional role of myostatin or GDF11 can be gleaned from studies of genetically modified mice. Genetic deficiency of GDF11 in mice causes profound developmental abnormalities, including agenesis of the kidneys and perinatal lethality [55,56]. The deletion of myostatin via gene targeting promoted both hypertrophy and hyperplasia of skeletal muscle, whereas postnatal inhibition of myostatin induced hypertrophy only [57]. In a study using double myostatin, GDF11 mutants demonstrated that myostatin and GDF11 are both required for limb development and axial skeletal development and patterning. The effect of deletion of both factors on the axial skeleton was greater than for the loss of GDF11 alone. These results provide evidence that that myostatin and GDF11 have redundant functions in regulating skeletal development [58].

The potential role of GDF11 on skeletal muscle and heart should be viewed with caution according to the various and major properties of myostatin on skeletal cells. As we reported, knock-out (KO) mouse studies suggest myostatin affects skeletal muscle mass and also cardiac growth. However, the cardiac consequences of inhibiting myostatin remain unclear. Myostatin is a potent negative regulator of skeletal muscle growth and its inactivation can induce skeletal muscle hypertrophy, while its overexpression or systemic administration causes muscle atrophy [59]. GDF11 and activins exert an inhibitory effect on myocyte differentiation, similar to that induced by myostatin, and their activity is neutralized by follistatin [60].

Many questions remain about the role of the extracellular matrix and stem cells on muscle development, and factors determining the early differentiation of myogenic cells. Induced pluripotent stem (iPS) cells can be differentiated into cardiomyocytes, suggesting that iPS cells have the potential to significantly advance future cardiac regenerative therapies. The TGF-β signaling pathway is one of the important regulators of pluripotency. In this context, it is suggested that a lack of myostatin epigenetically reprograms the muscle derived stem cells (MDSCs) to become pluripotent stem cells. MDSCs are multipotent stem cells that can differentiate into skeletal muscle precursor cells [61]. Favorable effects of stem cells on perfusion and myocardial blood flow have been reported in a number of trials of cell therapy. Combining different types of stem cells lead to superior outcome in eliciting cardiac repair. Emerging approaches, including genetic modification, stem cell-derived exosomes, and new pharmacological drugs, have been applied to improve the efficacy of stem cell therapy [62]. It will be important to investigate the mechanisms that induce the myogenic effects of GDF11 with therapeutic possibilities for a variety of age-related diseases in association with cell therapy.

## 8. Summary

At a mechanistic level, evidence now exists that the beneficial effects of young blood on cognitive performance and cardiovascular function may be mediated by various rejuvenating factors and that many of the pro-longevity signaling pathways play important roles in organ functions. An important set of molecular mediators of regenerative capacity are those implicated in the TGF- β pathway [63] and hormones such as oxytocin [64]. At the level of cellular mechanisms, systemic inhibition of the TGF-beta type1 receptor enhanced both neurogenesis and myogenesis and normalized beta-2 microglobulin (B2M). Systemic B2M accumulation in aging blood promotes age-related cognitive dysfunction and impairs neurogenesis [65]. Furthermore, studies suggested that B2M could be a biomarker for CAD, with B2M being a predictor of all-cause mortality and adverse cardiac events in patients [66].

In conclusion, HP and old plasma administration studies have demonstrated the existence of “pro-aging” factors, such as negative regulators of regenerative capacity in peripheral tissues and the brain, and that impaired regenerative capacity is associated with alterations in cardiovascular and cognitive functions [67]. It will also be important to determine whether long-term treatment with rGDF11 has any negative consequences, especially since GDF11 is known to regulate cell proliferation in the development of multiple organ systems. The existence of possible field cancerization should be taken into consideration.

## Figures and Tables

**Figure 1 ijms-19-03998-f001:**
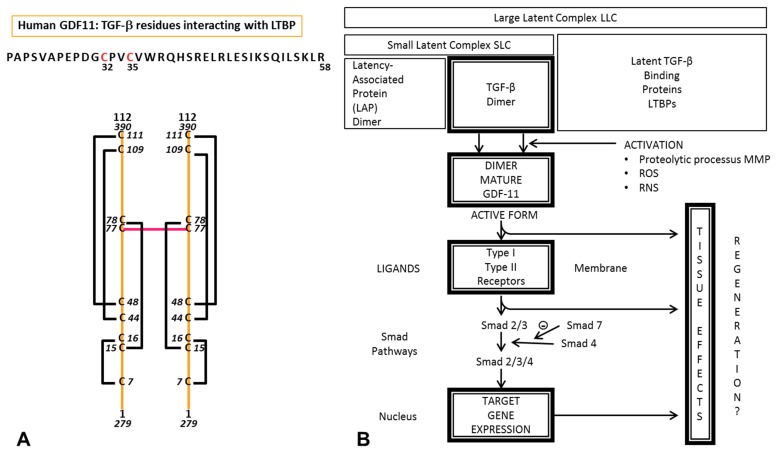
Structural organization of GDF11 mature domain (**A**) and schematic representation of GDF11 signaling pathways (**B**). (A) Mature GDF11 exists as a disulfide-linked homodimer with apparent molecular weights of 25 kDa. The active domain of GDF11 is typically found as a dimer, linked by cysteine-bonds. The symmetrical dimer forms two distinct interfaces for receptor binding. All the other cysteine C are involved in intrachain disulfide bonds. As show in the schematic representation, there are four disulfide bounds in each homodimer. C plays an important role in stabilization of protein structure The LTBPs (latent transforming growth factor β binding proteins) are components of the extracellular matrix, identified as forming latent complexes with TGF-β. (B) TGF-β is associated with latent transforming growth factor B – binding proteins (LTBPs) inside the cell form the large latent complex (LLC). TGF-β is secreted in a latent dimeric complex containing the C-terminal mature TGF-β and its N-terminal pro-domain, LAP (TGF-β latency associated protein). Proteolytic activation via matrix metalloproteinases (MMPs) and reactive oxygen species (ROS) are associated with the TGF-β activation mechanism, releasing an active dimer to elicit signaling. GDF11 (active form), a TGF-β signaling factor (407 amino acids) exerts its effects by binding to specific receptors. Members of TGF-β family bind to activin-receptor-like-kinase: ALK-Type I and Type II. ALK-I induces phosphorylation of Smads. Ligands of various TGF-β receptors lead to recruitment and activation of Smads transcription factors that regulate gene expression in nucleus. Differences in GDF11 expression and cellular effects vary depending on tissue localization.

**Figure 2 ijms-19-03998-f002:**
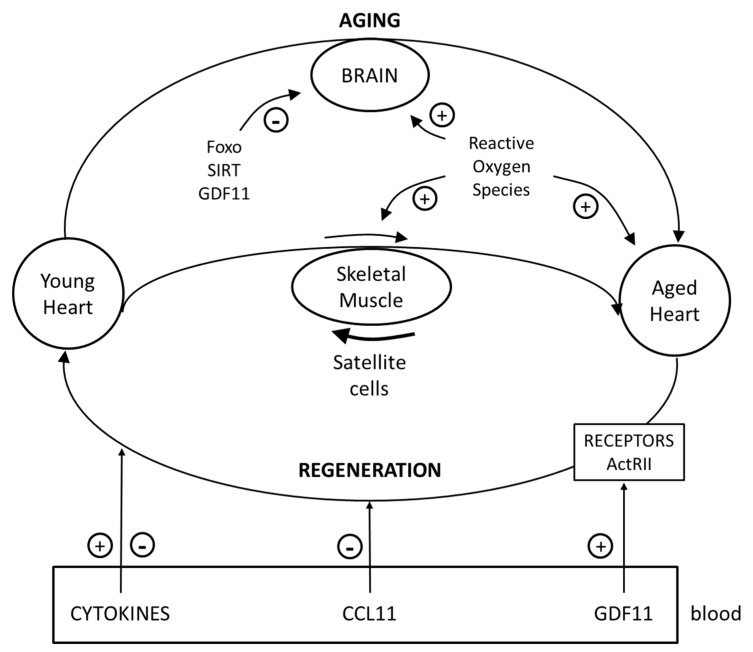
Age-related functional impairment in the heart, skeletal muscle, and brain—regenerative pathways. Systemic factors can affect aging-associated events, either positively or negatively. “Pro-youthful” factors, such as GDF11, present in young or aged blood may be able to improve cardiac regeneration. Another factor in blood is the chemokine (C-C motif) chemokine 11: CCL11. Age-related heart failure can be reversed by exposure to GDF11. During aging, there is a significant decline in skeletal muscle regenerative function and muscle satellite cells are responsible for the regeneration capacity of the muscle. Age-related mechanisms impair the function of the neurovascular unit in the brain. Foxo, SIRT, GDF11, and reactive oxygen species are modulators of the age-related process.

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
