# Peer review of "Regenerative Capacity of Endogenous Factor: Growth Differentiation Factor 11; a New Approach of the Management of Age-Related Cardiovascular Events"

_ijms, 2018, doi:10.3390/ijms19123998_

Round 1
Reviewer 1 Report
The manuscript by Rochette L et al. summarizes the properties of
endogenous factor growth differentation factor 11 (GDF11) in the management of age-related cardiovascular
events. The authors have made a goog job, because the review is interesting ,
clear and well written.
Author Response
The authors thank the Reviewer for the careful review of our manuscript and for the nice comments he/she made.
Reviewer’comment:
The manuscript by Rochette L et al. summarizes the properties of endogenous factor growth differentiation factor 11 (GDF11) in the management of age-related cardiovascular events. The authors have made a good job, because the review is interesting , clear and well written.
Reviewer 2 Report
This review article focused on growth differentiation factor 11 (GDF11) as a regenerative factor for various age-related diseases. It contains a lot of useful information and have an impact for the readers in the cardiovascular and regenerative research field. However, there are several points that have to be improved concerning this manuscript. These are given below.
1. Typographical errors are observed. The manuscript should be checked time after time with extreme caution.
2. The authors should use abbreviations throughout the manuscript after the initial use (e.g. CVD, HP).
3. Figure 1, especially 1A, should be explained in more detail.
4. The authors should insert some references in the third section “The surgical technique of parabiosis”.
5. The title should be reconsidered because this article focused not only on age-related cardiovascular events but also on impairment of skeletal muscle and brain.
6. Figures for section 4 (heart), 5 (skeletal muscle) and 6 (brain) at least section 4 about the efficacy of GDF11 should be added.
Author Response
The authors thank the Reviewer for the careful review of our manuscript and for the interesting suggestions he/she made.
Reviever’comments (red)
This review article focused on growth differentiation factor 11 (GDF11) as a regenerative factor for various age-related diseases. It contains a lot of useful information and have an impact for the readers in the cardiovascular and regenerative research field. However, there are several points that have to be improved concerning this manuscript.
In agreement with the Reviewer we modified our manuscript.
Typographical errors are observed. The manuscript should be checked time after time with extreme caution.. The authors should use abbreviations throughout the manuscript after the initial use (e.g. CVD, HP).
This new review has been redacted in order to correct the typographical errors and use the initial use of abbreviations in agreement with yours comments
Figure 1, especially 1A, should be explained in more detail.
As suggested, we added the following sentences:
Mature GDF11 exists as a disulfide-linked homodimer with apparent molecular weights of 25 kDa. The active domain of GDF11 is typically found as a dimer, linked by cysteine-bonds. The symmetrical dimer forms two distinct interfaces for receptor binding. All the other cysteine C are involved in intrachain disulfide bonds. As show in the schematic representation, there are four disulfide bounds in each homodimer. C plays an important role in stabilization of protein structure The LTBPs (latent transforming growth factor β binding proteins) are components of the extracellular matrix, identified as forming latent complexes with TGF-β
The authors should insert some references in the third section “The surgical technique of parabiosis”.
As suggested, we added 3 references in this section
Song G, Nguyen DT, Pietramaggiori G, Scherer S, Chen B, Zhan Q, Ogawa R, Yannas IV, Wagers AJ, Orgill DP, Murphy GF. Wound Repair Regen. 2010 Jul-Aug;18(4):426-32.
Parabiosis in mice: a detailed protocol. Kamran P, Sereti KI, Zhao P, Ali SR, Weissman IL, Ardehali R. J Vis Exp. 2013 Oct 6;(80).
Heterochronic parabiosis: historical perspective and methodological considerations for studies of aging and longevity. Conboy MJ, Conboy IM, Rando TA. Aging Cell. 2013 Jun;12(3):525-30.
The title should be reconsidered because this article focused not only on age-related cardiovascular events but also on impairment of skeletal muscle and brain.
We agree with the Reviewer comment that the title should be modified. We propose: Regenerative capacity of endogenous factor: GDF11; a new approach of the management of age-related cardiovascular events
Figures for section 4 (heart), 5 (skeletal muscle) and 6 (brain) at least section 4 about the efficacy of GDF11 should be added.
As suggested, we added a new figure in relationship with the age-related functional impairment in heart, skeletal muscle and brain and the regenerative pathways.
Systemic factors can affect aging-associated events, either positively or negatively. “Pro-youthful” factors such as GDF11 present in young or aged blood may be able to improve cardiac regeneration. Another factor in blood is the chemokine (C-C motif) chemokine 11: CCL11. Age-related heart failure can be reversed by exposure to GDF11. During aging, there is a significant decline in skeletal muscle regenerative function and muscle satellite cells are responsible for the regeneration capacity of the muscle. Age-related mechanisms impair the function of the neurovascular unit in the brain. Foxo, SIRT, GDF11, and reactive oxygen species are modulators of age-related process.